# Bacteraemia Is Associated with Increased ICU Mortality in the Postoperative Course of Lung Transplantation

**DOI:** 10.3390/antibiotics11101405

**Published:** 2022-10-13

**Authors:** Alexy Tran-Dinh, Marion Guiot, Sébastien Tanaka, Brice Lortat-Jacob, Enora Atchade, Nathalie Zappella, Pierre Mordant, Yves Castier, Hervé Mal, Gaelle Weisenburger, Jonathan Messika, Nathalie Grall, Philippe Montravers

**Affiliations:** 1Département d’Anesthésie-Réanimation, Hôpital Bichat Claude Bernard, AP-HP, Université Paris Cité, 75018 Paris, France; 2INSERM UMR 1148 LVTS, Université Paris Cité, 75018 Paris, France; 3INSERM U1188 Diabetes Atherothrombosis Réunion Indian OCean (DéTROI), CYROI Plateform, Réunion Island University, 97744 Saint-Denis de la Réunion, France; 4Service de Chirurgie Vasculaire, Thoracique et Transplantation Pulmonaire, Hôpital Bichat Claude Bernard, AP-HP, Université Paris Cité, 75018 Paris, France; 5INSERM UMR 1152 PHERE, Université Paris Cité, 75018 Paris, France; 6Pneumologie B et Transplantation Pulmonaire, Hôpital Bichat Claude Bernard, AP-HP, Université Paris Cité, 75018 Paris, France; 7Paris Transplant Group, 75018 Paris, France; 8Service de Bactériologie, Hôpital Bichat Claude Bernard, AP-HP, Université Paris Cité, 75018 Paris, France; 9INSERM UMR 1137 IAME, Université Paris Cité, 75018 Paris, France

**Keywords:** lung transplantation, bacteraemia, intensive care unit

## Abstract

We aimed to describe the prevalence, risk factors, morbidity and mortality associated with the occurrence of bacteraemia during the postoperative ICU stay after lung transplantation (LT). We conducted a retrospective single-centre study that included all consecutive patients who underwent LT between January 2015 and October 2021. We analysed all the blood cultures drawn during the postoperative ICU stay, as well as samples from suspected infectious sources in case of bacteraemia. Forty-six bacteria were isolated from 45 bacteraemic patients in 33/303 (10.9%) patients during the postoperative ICU stay. *Staphylococcus aureus* (17.8%) was the most frequent bacteria, followed by *Pseudomonas aeruginosa* (15.6%) and *Enterococcus faecium* (15.6%). Multidrug-resistant bacteria accounted for 8/46 (17.8%) of the isolates. The most common source of bacteraemia was pneumonia (38.3%). No pre- or intraoperative risk factor for bacteraemia was identified. Recipients who experienced bacteraemia required more renal replacement therapy, invasive mechanical ventilation, norepinephrine support, tracheotomy and more days of hospitalization during the ICU stay. After adjustment for age, sex, type of LT procedure and the need for intraoperative ECMO, the occurrence of bacteraemia was associated with a higher mortality rate in the ICU (aOR = 3.55, 95% CI [1.56–8.08], *p* = 0.003). Bacteraemia is a major source of concern for lung transplant recipients.

## 1. Introduction

Lung transplantation (LT) is a life-saving therapy for patients with end-stage lung disease. Posttransplant infections account for 17% and 33% of the causes of death between 0 and 30 days and between 31 days and 1 year, respectively [1]. Bacteraemia, defined as the presence and detection of bacteria in the blood, is one of the top ten causes of death in the United States and Europe [2]. Although bacteraemia has been shown to increase morbidity and mortality among solid organ transplant recipients [3], it remains insufficiently described in LT [4,5,6].

During the postoperative stay in the intensive care unit (ICU), lung transplant patients receive high doses of immunosuppressive drugs, putting them at high risk for infectious complications [7], and are exposed to many other complications [8,9]. Therefore, we aim to describe the prevalence, risk factors, morbidity and mortality associated with the occurrence of bacteraemia during the posttransplant ICU stay, which have not yet been reported.

## 2. Results

Between January 2015 and October 2021, 303 patients underwent LT. Demographic data, underlying diseases and indications for LT are presented in Table 1.

### 2.1. Prevalence of Bacteraemia and Characteristics

Forty-six bacteria were isolated from forty-five cases of bacteraemia occurring in 33/303 (10.9%) patients during the postoperative ICU stay. The median (IQR) time from LT to the onset of bacteraemia was 8 (1–23) days. The distribution of first episodes of bacteraemia according to Gram stain and time from LT is presented in Figure 1.

Monomicrobial bacteraemia (*n* = 44) was observed in 32/33 (97.8%) patients (one patient had a mixed *Staphylococcus aureus* and *Proteus mirabilis* bacteraemia). *Staphylococcus aureus* (17.8%) was the most frequent bacteria, followed by *Pseudomonas aeruginosa* (15.6%) and *Enterococcus faecium* (15.6%). MDR bacteria accounted for 8/46 (17.8%) of the isolates, exclusively involving Gram-negative bacilli (GNB), with the exception of *coagulase-negative staphylococci*. MDR GNB included multidrug-resistant *P. aeruginosa* (resistance to ceftazidime, cefepime, ciprofloxacin, meropenem) (*n* = 1), extended-spectrum β-lactamase (ESBL)-producing *Enterobacter cloacae* (*n* = 1), ESBL-producing *Klebsiella pneumonia* (*n* = 3), ESBL-producing *Escherichia coli* (*n* = 2) and AmpC β-lactamase-overproducing *Klebsiella aerogenes* (*n* = 1). No methicillin-resistant *S. aureus* strains were isolated.

The most common source of bacteraemia was pneumonia (38.3%). However, the sources remained unknown in 31.2% of cases. Some cases of bacteraemia had multiple sources. One case due to *K. pneumonia* and one due to *E. faecium* both had two sources (pneumonia and pleural infection and chest wall infection and pleural infection, respectively). The microbiological characteristics are displayed in Table 2. One patient experienced a recurrence of bacteraemia during the posttransplant ICU stay. The first incidence of bacteraemia was identified as *P. aeruginosa* and was due to a chest wall infection 33 days after LT. A recurrence due to pleural infection occurred at day 68. Only one *Candida-albicans*-positive blood culture of unknown origin was observed during the postoperative ICU stay, drawn on ICU admission.

### 2.2. Pre-Existing Risk Factors at ICU Admission Associated with the Occurrence of Bacteraemia during the Postoperative ICU Stay

No pre-existing risk factors at ICU admission were associated with the occurrence of bacteraemia during the postoperative ICU stay (Table 1).

### 2.3. ICU Morbidity Associated with the Occurrence of Bacteraemia

Recipients who experienced bacteraemia had higher SAPS II scores on postoperative ICU admission. In addition, during their ICU stays, they had higher acute kidney injury KDIGO scores, required more renal replacement therapy, invasive mechanical ventilation, norepinephrine support, tracheotomy and had longer ICU stays (Table 3).

### 2.4. ICU and One-Year Mortality Rates Associated with the Occurrence of Bacteraemia

A total of 36.4% of recipients with postoperative bacteraemia died in the ICU compared to 12.6% for patients without bacteraemia (*p* < 0.001). After adjustment for age, sex, type of LT procedure and the need for intraoperative ECMO, the occurrence of bacteraemia was associated with a higher mortality rate in the ICU (aOR = 3.55, 95% CI (1.56–8.08), *p* = 0.003).

The 1-year survival was inferior for recipients who had bacteraemia occurring during the postoperative ICU stay (57.6%) compared to that of recipients without bacteraemia (77.4%). The Kaplan–Meier survival curve and log-rank test (*p* = 0.03) are shown in Figure 2. After adjustment for age, sex, type of LT and the need for ECMO, the difference in the incidence of bacteraemia did not reach statistical significance to independently decrease 1-year survival (HR = 1.70, 95% CI (0.93–3.09), *p* = 0.09).

### 2.5. Impact of the Adequacy of 48 h Perioperative ABX to Bacteraemia Occurring during the First Week after LT on ICU Morbidity and Mortality

Sixteen (16/33 = 48.5%) patients had bacteraemia within the first week of LT. Forty-eight-hour perioperative ABX had effective in vitro activity against the bacterial strains of bacteraemia occurring during the first week in 11 patients (amoxicillin/clavulanic acid (*n* = 6), cefazolin (*n* = 2), piperacillin/tazobactam (*n* = 2), cefepime (*n* = 1)) and was ineffective in 5 patients (amoxicillin/clavulanic acid (*n* = 2), cefazolin (*n* = 2), piperacillin/tazobactam + ciprofloxacin (*n* = 1)). Patients with and without effective 48 h perioperative ABX had similar rates of ICU mortality (60% vs. 27.3%, *p* = 0.30), length of ICU stay (26 (10–51) vs. 16 (7–69) days, *p* = 1), duration of invasive mechanical ventilation (25 (10–27) vs. 3 (1–28) days, *p* = 0.42) and duration of norepinephrine support (9 (7–10) vs. 2 (1–9) days, *p* = 0.30).

### 2.6. Comparison of ICU Morbidity and Mortality between Patients with Postoperative Bacteraemia Occurrence before 7 Days vs. after 7 Days

The rate of ICU mortality was similar between patients with postoperative bacteraemia occurrence before days vs. after 7 days (37.5% vs. 35.3%, *p* = 0.50). However, patients who developed bacteraemia early (before 7 days) had a longer duration of invasive mechanical ventilation (8 (1–31) vs. 38 (32–55) days, *p* = 0.01) and ICU length of stay (18 (7–60) vs. 55 (41–96) days, *p* = 0.01) than patients who developed bacteraemia after 7 days. The duration of norepinephrine support did not significantly differ between the groups (3 (1–10) vs. 4 (2–19) days, *p* = 0.4)

### 2.7. Comparison of ICU Morbidity and Mortality between Patients with Gram-Positive vs. Gram-Negative Postoperative Bacteraemia

Patients with Gram-positive bacteraemia had a similar rate of ICU mortality (27.8% vs. 43.8%, *p* = 0.15), duration of invasive mechanical ventilation (33 (13–49) vs. 32 (2–44) days, *p* = 0.65), norepinephrine support (6 (2–16) vs. 3 (1–15) days, *p* = 1) and ICU length of stay (45 (22–96) vs. 44 (22–89) days, *p* = 0.91) as patients with Gram-negative bacteraemia.

### 2.8. Impact of Multidrug-Resistant Isolates on ICU Mortality of Recipients with Bacteraemia

Patients with MDR bacteraemia had a similar rate of ICU mortality (37.5% vs. 36%, *p* = 1), duration of invasive mechanical ventilation (49 (35–55) vs. 22 (3–40) days, *p* = 0.06), norepinephrine support (7 (3–12) vs. 3 (1–14) days, *p* = 0.61) and ICU length of stay (95 (48–121) vs. 37 (16–56) days, *p* = 0.08) as patients with non-MDR bacteraemia.

## 3. Discussion

This study examined the largest cohort of lung transplant recipients studied for bacteraemia, on par with that of Husain et al. [4]. However, significant differences exist. Our cohort did not include patients with cystic fibrosis and primarily consisted of patients with COPD and ILD, and we focused on describing bacteraemia and its impact on postoperative ICU stay. Only three studies published more than 15 years ago have examined the impact of bacteraemia in LT [4,5,6]. These studies reported rates of bacteraemia ranging from 11.5% to 25%, higher than our observed rate of 10.5%. This difference could be explained by the absence of cystic fibrosis cases in our cohort, a risk factor for bacteraemia that has already been identified [5]. In addition, we focused only on bacteraemia occurring during the postoperative ICU stay.

Bacteraemia has been consistently associated with decreased survival in lung transplant recipients. Here, we showed that bacteraemia drastically increased early morbidity and mortality during the postoperative ICU stay and could impact 1-year survival. As in critically ill nontransplant patients in whom bacteraemia is associated with organ failure and mortality [10,11], this complication could echo the severity of a patient’s condition. In the setting of LT, recipients are exposed to major risk factors for bacteraemia, namely ICU hospitalization, immunosuppression, nosocomial pneumonia and the need for mechanical ventilation. The rate of bacteraemia observed here was, consequently, two to five times more frequent than in nontransplant patients [10,11]. The high mortality associated with bacteraemia may be related to the fact that one-third of the bacteraemia incidents had an unknown source, which may make it more difficult to treat. In addition, bacteraemia has previously been shown to be an independent risk factor for mortality in nosocomial pneumonia [12], which was the most common source in our cohort.

In terms of public health, patients with bacteraemia required more invasive mechanical ventilation, renal replacement therapy and a longer length of ICU stay, which are very costly. Nosocomial bloodstream infections in critically ill patients were already shown to cause a significant economic burden [11].

Bacteraemia was most commonly due to *S. aureus*, *P. aeruginosa*, and *E. faecium,* the latter being found at a somewhat surprising level compared with those reported in previous studies [4,5,6]. However, in two-thirds of the cases, the source was not identified. MDR isolates were found only among GNB and predominantly involved ESBL-producing *Enterobacterales*. Neither MDR nor GNB were associated with higher mortality in recipients with bacteraemia. The rate of 17.8% MDR documented in the isolates was lower than the 48% rate reported by Husain et al. However, in their study, MDR involved all the *Burkholderia cepacia* isolates and 75% of the *S. aureus* strains, whereas in our cohort, *B. cepacia* was never found, as cystic fibrosis was not included. Furthermore, methicillin-resistant *Staphylococcus aureus* was not isolated, which is consistent with its higher prevalence in the US than in Europe [13,14,15].

Bacteraemia is likely a surrogate marker of severity, but unfortunately, we were unable to determine any preexisting factors at postoperative ICU admission that were associated with the occurrence of bacteraemia to identify a subpopulation of at-risk patients. Fifty percent of the bacteraemia cases occured early, within 8 postoperative days, despite the absence of induction therapy by T-cell depletion, which has been identified as a risk factor [4]. Cystic fibrosis and mechanical ventilation before LT have also been identified as risk factors for bacteraemia [5]. However, we could not evaluate these variables because we did not perform LT under these conditions. The lack of impact of adequate antibiotic prophylaxis on the occurrence of bacteraemia in the first week after LT was particularly disappointing. This lack of effect may be related to the fact that curative antibiotic therapy initiated upon the detection of bacteraemia was sufficient to treat it effectively, which was reinforced by the observation of a single recurrent bacteraemia related to pleural infection after chest wall infection. The emergence of new rapid molecular and phenotypic tests for the identification and susceptibility profile of positive blood cultures could further reduce the time to effective treatment and the misuse of antibiotics [16,17,18]. Additional biomarkers could also be of interest, such as procalcitonin, for the early detection of a quick ascent of an inflammation surrogate that should decrease after surgery [19]. In the future, point-of-care diagnostics could also be expected to accelerate the therapeutic process.

This study had some limitations related to its retrospective and monocentric nature with a limited number of cases of bacteraemia, which calls for great caution in interpreting the data and generalizing the results.

## 4. Materials and Methods

### 4.1. Study Design

We conducted a retrospective single-centre study that included all consecutive patients who underwent LT between January 2015 and October 2021. Retransplantations and ex vivo lung perfusion procedures were not included.

The study was conducted in accordance with the Declaration of Helsinki and approved by the ethics committee of CEERB Paris Nord, which waived the need for signed informed consent (Institutional Review Board -IRB 00006477- Université Paris Cité, AP-HP.Nord).

We analysed all blood cultures drawn during the postoperative ICU stay, as well as samples from potential infectious sources in case of bacteraemia, defined as sites suspected of being infected by and responsible for bacteraemia.

The primary objectives were to describe (i) the prevalence of bacteraemia in the postoperative ICU stay, (ii) the bacterial species involved and their antibiotic susceptibility profiles, (iii) the potential pre- and perioperative risk factors for bacteraemia and (iv) the consequences of bacteraemia in terms of outcome (ICU morbidity and mortality).

As exploratory outcomes, we assessed (i) the impact of the adequacy of 48 h perioperative antibiotic prophylaxis (ABX) on bacteraemia occurring during the first week after LT morbidity and mortality in the ICU, (ii) the comparison of morbidity and mortality in the ICU between patients with postoperative bacteraemia occurrence before 7 days vs. after 7 days, (iii) the comparison of morbidity and mortality in the ICU between patients with Gram-positive vs. Gram-negative postoperative bacteraemia and (iv) the impact of multidrug-resistant (MDR) bacteria on morbidity and mortality in the ICU for recipients with bacteraemia.

### 4.2. Microbiological Features and Definitions

Each blood culture set included two bottles (aerobic and anaerobic). Ten millilitres of blood was taken under sterile conditions and injected into each bottle. All bottles were handled using a BD BACTEC™ blood culture system (Becton-Dickinson), which gave an automatic computerized alert when bacterial growth was detected and recorded the time elapsed between the introduction of the bottle in the apparatus and the alert signal. Bottles were incubated for 5 days at 35 °C (or extended to 8 days if infective endocarditis was suspected).

The diagnostic procedures conducted for all positive bottles included Gram staining, bacterial isolation using standard bacteriologic techniques and bacterial identification at the species level with a matrix-assisted laser desorption ionization-time of flight mass spectrometry (MALDI-TOF MS) Microflex LT Biotyper (Bruker Daltonics, Bremen, Germany). Bacterial susceptibility to antibiotics was determined using the disk-diffusion method according to EUCAST [20,21].

Bacteraemia was defined as the isolation of bacteria from a blood culture in the presence of clinical signs of infection, according to the Centers for Disease Control and Prevention criteria (central-line-associated bloodstream infection (CLABSI) and noncentral-line-associated bloodstream infection) [22].

For the usual bacteria considered to be skin contaminants (coagulase-negative *staphylococci* (CNS), *Bacillus* spp., *Corynebacterium* spp. or *Cutibacterium acnes*), bacteraemia was considered if more than two positive blood cultures were identified in a 3-day period [23].

The source of the bacteraemia (pneumonia, pleural infection, chest wall infection, intra-abdominal infection or other) was defined as a site sampled with a positive microbiological culture, according to the infection threshold where applicable, with the same bacterial species and the same antibiotic susceptibility as those isolated from the blood culture. If no source was identified, the bacteraemia was classified as being of an unknown source.

Recurrent bacteraemia was defined as the occurrence of a subsequent episode of the same microorganism with the same antibiotic susceptibility profile in a blood culture during the study period at least one week after the resolution of a previous episode (based on the resolution of symptoms and subsequent negative blood cultures with an appropriate duration of antibiotic therapy and appropriate source control, if indicated) [24].

Patients with bacteraemia were treated with appropriate antibiotic therapy, i.e., bacteria were susceptible according to the antibiotic susceptibility testing, for a duration depending on the infectious source, combined with source control if indicated: 7 days for pneumonia, 21 days for pleural infection, 7 days for catheter-related infection, 7 days for intra-abdominal infection, 7 days for chest wall infection and 6 weeks for infective endocarditis.

### 4.3. Data Collection

We collected:All positive blood cultures drawn during the posttransplant ICU stay; the time from LT to onset of bacteraemia, the type of bacterial species and the presence of MDR profiles were recorded [25].The demographic and pre-existing characteristics of patients before postoperative ICU admission, including the following: age, sex, body mass index (BMI), primary diagnosis of chronic pulmonary disease, cytomegalovirus mismatch (recipient/donor+), past medical history of diabetes and revascularized ischaemic heart disease, high-emergency LT, extracorporeal membrane oxygenation (ECMO) as a bridge to transplant and mean pulmonary arterial pressure (mPAP) measured by a right-heart catheterization at listing. High-emergency LT is a national prioritization system for the most severe patients with fibrosis, cystic fibrosis or pulmonary hypertension that was introduced in France in 2007 [26].Intraoperative characteristics: type of LT (i.e., single or bilateral), maximum graft cold ischaemic time, intraoperative blood transfusion of more than three packed red blood cells (PRBC) and intraoperative ECMO.Postoperative outcomes in ICU: simplified acute physiology score II (SAPS II) and sequential organ failure assessment (SOFA) score at ICU admission, acute kidney injury stage 3 of KDIGO (Kidney Disease: Improving Global Outcomes), renal replacement therapy, duration of mechanical ventilation, duration of norepinephrine support, ECMO in ICU, tracheotomy, ICU length of stay and mortality rates at day 30 and 1 year.

### 4.4. Perioperative Management

Surgical transplantation procedures and perioperative care, including postoperative management, were standardized for all patients according to our local protocol already published elsewhere [19]. The immunosuppressive regimen included mycophenolate mofetil, corticosteroids and tacrolimus. Induction therapy was not performed.

Perioperative antibiotic prophylaxis (ABX) was defined by the antibiotic regimen started intraoperatively. The standard perioperative ABX was cefazolin, as recommended in “Clinical practice guidelines for antimicrobial prophylaxis in surgery” [27]. However, ABX was modified for 1) antibiotic therapy given to the donor before lung harvesting and 2) adapted antibiotic therapy in cases of bronchoalveolar colonization ≤ 3 months documented with cefazolin-resistant bacteria. ABX was started intraoperatively and continued for 48 h after surgery, as recommended [28]. Antibiotic therapy was adapted to microbiological cultures obtained from bronchoalveolar lavage (BAL) performed on postoperative ICU admission. If a BAL culture was negative without evidence of infection, ABX was stopped after 48 h.

### 4.5. Statistical Analysis

The baseline characteristics within each group were described with numbers and percentages for qualitative variables and with medians and interquartile ranges for quantitative variables.

Univariate logistic regression and a calculation of unadjusted odds ratios (ORs) and 95% CIs were used to assess (1) risk factors pre-existing to ICU admission and associated with bacteraemia, (2) postoperative outcomes in the ICU associated with bacteraemia, (3) the impact of the adequacy of 48 h perioperative ABX to bacteraemia occurring during the first week after LT on morbidity and mortality in the ICU, (4) the comparison of morbidity and mortality in the ICU between patients with postoperative bacteraemia occurrence before 7 days vs. after 7 days, (5) the comparison of morbidity and mortality in the ICU between patients with Gram-positive vs. Gram-negative postoperative bacteraemia and (6) the impact of multidrug resistance on morbidity and mortality in the ICU for bacteraemic recipients.

The impact of bacteraemia on ICU mortality was assessed by multivariate logistic regression adjusted for age, sex, type of LT procedure and intraoperative ECMO to calculate adjusted OR (aOR) and 95% CI.

To analyse the impact of bacteraemia on 1-year survival, the probability of all-cause death was estimated using the Kaplan–Meier method and compared with the log-rank test. The risk of 1-year all-cause mortality was estimated before and after adjustment using the Cox proportional hazards model. We selected clinically relevant covariates analysed in a univariate regression to fit the multivariate method. The results were expressed as hazard ratios (HRs) and 95% confidence intervals (95% CIs).

All reported *p*-values were two-sided, and the level of statistical significance was specified a priori as less than 0.05. Statistical analysis and data management were performed using BM SPSS Statistics version 20 (IBM Corp., Armonk, NY, USA)

## 5. Conclusions

Bacteraemia is a major source of concern for lung transplant recipients that has been reported in more than 10% of these patients during the postoperative ICU stay and decreases their survival both in the ICU and at 1 year. As no risk factors have been identified, all recipients should be considered at risk. Future studies seem essential to evaluate the efficacy of the early detection of bacteraemia, including studies with rapid microbiologic diagnostic tests or biomarkers.

## Figures and Tables

**Figure 1 antibiotics-11-01405-f001:**
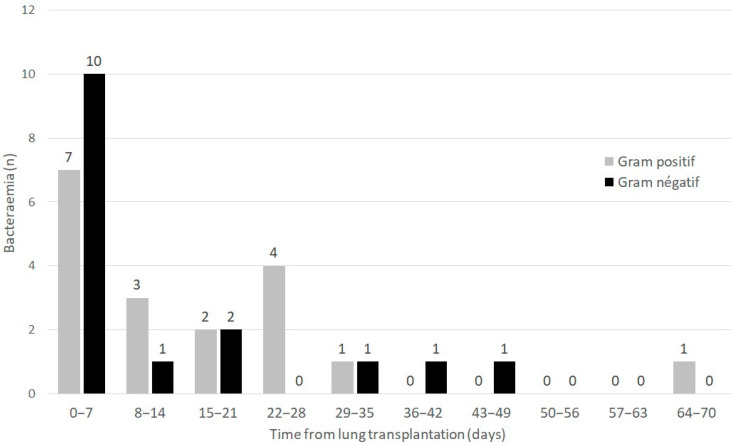
Distribution of first episodes of bacteraemia according to Gram stain and time from lung transplantation.

**Figure 2 antibiotics-11-01405-f002:**
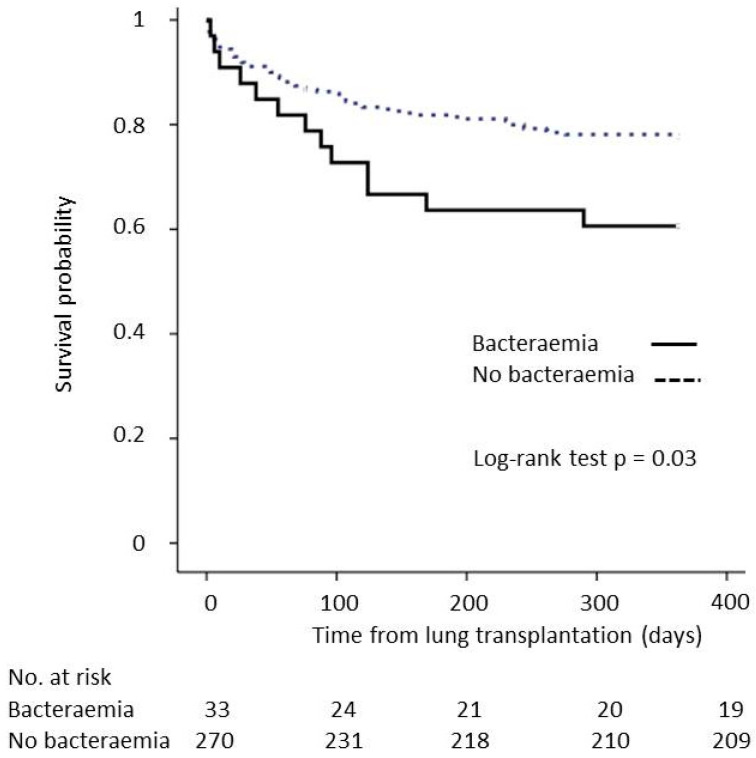
Impact of bacteraemia occurring during the posttransplant ICU stay on 1-year survival.

**Table 1 antibiotics-11-01405-t001:** Comparison of demographics and preoperative characteristics between lung transplant recipients with and without bacterial bloodstream infection.

	Total Patients (*n* = 303)	No Bacteraemia (*n* = 270)	Bacteraemia (*n* = 33)	Univariate Analysis OR [CI 95%]	*p*
Demographics and preoperative characteristics					
Age, years	57 (51–63)	57 (51–63)	55 (47–58)	0.98 (0.95–1.01)	0.13
Male sex	197 (65%)	176 (65.2)	21 (63.6)	0.94 (0.44–1.98)	0.86
BMI, kg/m^2^	24 (20–27)	24 (20–27)	26 (22–29)	1.07 (0.98–1.16)	0.12
Primary diagnosis					
COPD	105 (34.7)	98 (36.3)	7 (21.2)	0.47 (0.20–1.13)	0.09
ILD	153 (50.5)	133 (49.3)	20 (60.6)	1.59 (0.76–3.31)	0.22
Other	46 (15.2)	40 (14.8)	6 (18.2)	1.28 (0.50–3.29)	0.61
Revascularized ischaemic heart disease	17 (5.6)	16 (5.9)	1 (3.0)	0.50 (0.06–3.87)	0.71
Diabetes	34 (11.2)	29 (10.7)	5 (15.2)	1.48 (0.53–4.14)	0.45
mPAP, mmHg	25 (20–30)	25 (20–30)	24 (17–29)	0.96 (0.95–1.05)	0.84
Mismatch CMV (R−/D+)	57 (18.8)	50 (18.5)	7 (21.2)	1.19 (0.49–2.88)	0.78
Preoperative ECMO	23 (7.6)	19 (7.0)	4 (12.1)	1.82 (0.58–5.72)	0.30
High-emergency LT	54 (17.8)	48 (17.8)	6 (18.2)	1.03 (0.40–2.63)	0.96
Intraoperative characteristics					
Double LT	202 (66.7)	178 (65.9)	24 (72.7)	1.38 (0.62–3.09)	0.43
Thoracic epidural analgesia	170 (56.1)	154 (57.0)	16 (48.5)	0.71 (0.34–1.46)	0.35
Maximum cold graft ischaemic time, min	330 (270–400)	335 (270–400)	330 (260–410)	1.01 (0.98–1.05)	0.43
Intraoperative ECMO	211 (69.6)	184 (68.1)	27 (81.8)	2.10 (0.84–5.28)	0.11
Transfusion ≥ 3 PRBCs	138 (45.5)	120 (44.4)	18 (54.5)	1.50 (0.73–3.10)	0.27

Qualitative and quantitative variables are represented as numbers (percentages) and medians [interquartile ranges], respectively. Abbreviations: BMI: body mass index; COPD: chronic obstructive pulmonary disease; ILD: interstitial lung disease; mPAP: mean pulmonary arterial pressure; CMV: cytomegalovirus; R−/D+: CMV serologic status for recipient and donor; ECMO: extracorporeal membrane oxygenation; LT: lung transplantation.

**Table 2 antibiotics-11-01405-t002:** Bacteria identified in positive blood cultures stratified by the source.

	Pneumonia	Pleural Infection	Chest Wall Infection	Intra-Abdominal Infection	Other	Unknown	Total (Bacteria)	
Gram-negative bacilli								
Non-fermenting GNB								
*Pseudomonas aeruginosa*	4	1	1	-	1	-	7	
*Pseudomonas putida*	-	-	-	-	-	1	1	
*Stenotrophomonas maltophilia*	1	-	-	-	-	-	1	
*Achromobacter xylosoxidans*	1	-	-	-	-	-	1	
*Enterobacterales*								
*Enterobacter cloacae*	1	-	-	-	1	-	2	
*Klebsiella pneumoniae*	3	1	-	-	-	-	3	
*Escherichia coli*	-	1	-	1	-	-	2	
*Proteus mirabilis*	1	-	-	-	-	1	2	
*Klebsiella aerogenes*	1	-	-	-	-	1	2	
Anaerobes								
*Bacteroides thetaiotaomicron*	-	-	-	-	-	1	1	
Gram-positive cocci								
*Staphylococcus aureus*	6	-	-	-	1	1	8	
*Staphylococcus epidermidis*	-	1	-	1	-	-	2	
*Staphylococcus haemolyticus*	-	-	-	-	-	2	2	
*Staphylococcus lugdunensis*	1	-	-	-	-	-	1	
*Enterococcus faecium*	-	1	1	1	-	5	7	
*Enterococcus faecalis*	-	-	-	1	-	3	4	
Total (source)	18	5	2	4	3	15		46
							47	

**Table 3 antibiotics-11-01405-t003:** Comparison of morbidity and mortality during the posttransplant ICU stay between patients with and without bacteraemia.

	Total Patients (*n* = 303)	No Bacteraemia (*n* = 270)	Bacteraemia (*n* = 33)	Univariate Analysis OR [CI 95%]	*p*
At ICU admission					
SAPS II	43 (38–50)	43 (38–50)	45 (42–55)	1.03 (1–1.05)	0.04
SOFA score	7 (6–9)	7 (6–9)	8 (6–10)	1.14 (0.98–1.33)	0.08
During the ICU stay					
AKI stage 3 of KDIGO	48 (15.8)	37 (13.7)	11 (33.3)	3.15 (1.41–7.03)	0.004
RRT	38 (12.5)	29 (10.7)	9 (27.3)	3.12 (1.32–7.35)	0.007
Duration of mechanical ventilation, days	3 (1–21)	3 (1–13)	30 (3–49)	1.02 (1.01–1.03)	<0.001
Duration of noradrenaline support, days	2 (1–4)	2 (1–4)	4 (1–13)	1.05 (1.01–1.09)	0.01
ECMO in ICU	83 (27.4)	70 (25.9)	13 (39.3)	1.86 (0.88–3.93)	0.10
Tracheotomy	82 (27.1)	59 (21.9)	23 (69.7)	8.26 (3.71–18.24)	<0.001
ICU length of stay, days	17 (11–35)	16 (11–28)	42 (20–88)	1.03 (1.02–1.04)	<0.001
ICU mortality	46 (15.2)	34 (12.6)	12 (36.4)	3.97 (1.79–8.79)	<0.001

Qualitative and quantitative variables are represented as numbers (percentages) and medians [interquartile ranges], respectively. Abbreviations: SOFA: sequential organ failure assessment; SAPS II: simplified acute physiologic score; ICU: intensive care unit; AKI: acute kidney injury; KDIGO: kidney disease: improving global outcomes; RRT: renal replacement therapy; ECMO: extracorporeal membrane oxygenation.

## Data Availability

The data presented in this study are available on request from the corresponding author.

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
