# Peer review of "Bacteraemia Is Associated with Increased ICU Mortality in the Postoperative Course of Lung Transplantation"

_antibiotics, 2022, doi:10.3390/antibiotics11101405_

Round 1

Reviewer 1 Report

The manuscript by Alexy Tran Dinh et al. presents a retrospective study of bacteremia occurrence in the patients following lung transplantation. The authors aimed to evaluate the prevalence, risk factors, morbidity and mortality of the patients.

Although overall, the manuscript is well prepared, in my opinion, it is a bit too cryptic for a broad readership. It would help to elaborate descriptions and explanations of the results and the conclusions. For example, the section on preexisting risk factors only offers a conclusion and refers to Table 1 without explanations. The authors conclude that SAPS II score for patients with bacteremia is higher, 45 vs 43. How significant in a clinical sense is such difference?

Since the journal is focused on antibiotics, more details would be appreciated that are related to antibiotic treatments. For example, since there were some differences in perioperative antibiotic treatments, were there any correlations between these antibiotic treatments and occurrence of bacteremia? This is when the patients who had bacteremia are compared to those who did not.

How was MDR defined?

Since the authors make a comment about the differences in S.a. occurrence between USA and Europe, providing the general geographic location of the patients would be useful.

It would help to elaborate the methods used to detect and identify bacteria.

Other comments:

  1. L. 25 Explain what sources
  2. L. 26, 64 explain the relationship between these 45 patients with bacteremia and 33 patients. How can it be 45 cases in 33 patients?
  3. L. 48 and elsewhere, multiple references shall be shown in one set of brackets.
  4. Table 1, the category “others” shall be listed last in the group of diagnosis
  5. L.85 No need to capitalize “Day”
  6. L. 138 How was the in vitro activity measured?
  7. L.182 Severity shall not relate to patients, but their condition.
  8. L.189 What is this number (21324159)?
  9. L. 197 The “previous studies” need references.
  10. L. 237 missed (ii)
  11. L.270 More than 2 cultures out of how many tested?
  12. L.282 The appropriate antibiotic therapy shall be explained here.

Author Response

We warmly thank the reviewer for his time in evaluating our manuscript and for the insightful comments.

Reviewer 2 Report

To the authors, 

Dinh et al. conducted a retrospective study titled “Bacteremia is associated with increased ICU mortality in the postoperative course of lung transplantation.”. The authors present an interesting manuscript, in which they describe the prevalence, risk factors, morbidity and mortality associated with the occurrence of bacteremia during the postoperative ICU stay after lung transplantation.

However, the article has also several shortcomings:

-       To identify the source of bacteremia, the authors state that a sampling site must have a positive microbiological culture with the same bacterial species and antibiotic susceptibility as those isolated from the blood culture. If no source was identified, bacteremia was classified as being of unknown source. How were the sampling sites chosen? How was sampling conducted? The authors state that BALs were performed upon postoperative ICU admission. The high frequency of pulmonary sampling may therefore have resulted in pneumonia being the most common cause of bacteremia.

-       Lines: 234-235: How do the authors define “potential infectious sources in case of bacteremia”? Were all microbiological samples available shortly after blood culture collection considered?

-       Lines 264-267: The authors state that in addition to isolation of bacteria from blood cultures, the presence of clinical signs of infection were necessary. How were clinical signs of infection assessed retrospectively? 

-       The authors may consider reporting the total number of cultures taken and the % of positive results. 

Minor comments:

-       Line 189: “(21324159)

-       Lines 240-249: The authors may consider listing exploratory outcomes as done for primary objectives as opposed to bullet points. 

Lines 263: Could the authors include a more precise reference to the disk-diffusion method according to EUCAST.

Author Response

We warmly thank the reviewer for his time in evaluating our manuscript and for theinsightful comments.

Reviewer 3 Report

  Here I present the review of the paper entitled “Bacteraemia is associated with increased ICU mortality in the postoperative course of lung transplantation.” submitted to Antibiotics. 

Paper investigates the impact of bacteriemia on clinical outcome in patients after lung transplant.

In my opinion, paper fits the journal aim and scope. 

Paper is written clearly, and language quality is good, there are no problems in understanding the text of the paper. The structure of the paper is also appropriate. 

Novelty of the paper is questionable, as several reports examined similar topic. 

Scientific quality of the paper is acceptbale. Authors constructed good hypothesis and proved it with logically planed series of experiments. ‘Abstract’, ‘introduction’, ‘results and discussion’ and ‘conclusion’ sections are written clearly. ‘Introduction’ is effective, clear and well organized. ‘Materials and method’ section provides necessary information. ‘Results’ are reported realty. ‘Discussion; is written in logical with the analyzed data. Conclusions are not exaggerated and based on the experimental results. Figures are good quality. Authors cited relevant and recently published papers to sum up, my overall assessment of the paper is fair.

Author Response

(The authors gave the same response as above.)

Round 2

Reviewer 2 Report

The authors addressed all my concerns. No further comments form my side.